# Cucurbitacins as Potent Chemo-Preventive Agents: Mechanistic Insight and Recent Trends

**DOI:** 10.3390/biom13010057

**Published:** 2022-12-27

**Authors:** Hardeep Singh Tuli, Prangya Rath, Abhishek Chauhan, Anuj Ranjan, Seema Ramniwas, Katrin Sak, Diwakar Aggarwal, Manoj Kumar, Kuldeep Dhama, E Hui Clarissa Lee, Kenneth Chun-Yong Yap, Sharah Mae Capinpin, Alan Prem Kumar

**Affiliations:** 1Department of Biotechnology, Maharishi Markandeshwar Engineering College, Maharishi Markandeshwar (Deemed to be University), Mullana-Ambala 133207, India; 2Amity Institute of Environmental Sciences, Amity University, Noida 201303, India; 3Amity Institute of Environmental Toxicology, Safety and Management, Amity University, Noida 201303, India; 4Academy of Biology and Biotechnology, Southern Federal University, 344090 Rostov-on-Don, Russia; 5University Centre for Research and Development, University Institute of Pharmaceutical Sciences, Chandigarh University, Mohali 140413, India; 6NGO Praeventio, 50407 Tartu, Estonia; 7Department of Chemistry, Maharishi Markandeshwar University Sadopur, Ambala 134007, India; 8Division of Pathology, ICAR-Indian Veterinary Research Institute, Bareilly 243122, India; 9Department of Pharmacology, Yong Loo Lin School of Medicine, National University of Singapore, Singapore 117600, Singapore; 10NUS Centre for Cancer Research (N2CR), Yong Loo Lin School of Medicine, National University of Singapore, Singapore 119077, Singapore

**Keywords:** cucurbitacins, anti-proliferation, apoptotic, anti-angiogenic, anti-metastatic, synergism

## Abstract

Cucurbitacins constitute a group of cucumber-derived dietary lipids, highly oxidized tetracyclic triterpenoids, with potential medical uses. These compounds are known to interact with a variety of recognized cellular targets to impede the growth of cancer cells. Accumulating evidence has suggested that inhibition of tumor cell growth via induction of apoptosis, cell-cycle arrest, anti-metastasis and anti-angiogenesis are major promising chemo-preventive actions of cucurbitacins. Cucurbitacins may be a potential choice for investigations of synergism with other drugs to reverse cancer cells’ treatment resistance. The detailed molecular mechanisms underlying these effects include interactions between cucurbitacins and numerous cellular targets (Bcl-2/Bax, caspases, STAT3, cyclins, NF-κB, COX-2, MMP-9, VEGF/R, etc.) as well as control of a variety of intracellular signal transduction pathways. The current study is focused on the efforts undertaken to find possible molecular targets for cucurbitacins in suppressing diverse malignant processes. The review is distinctive since it presents all potential molecular targets of cucurbitacins in cancer on one common podium.

## 1. Introduction

Cancer is a huge global threat, representing a heavy burden to social systems and health care sectors all over the world. Moreover, over the past decades, both the incidence as well as mortality rates of malignant disorders have been increased, with a continuous sharp rise expected for further years [1]. This situation clearly indicates that current therapeutic tools for combating cancer are insufficient, inducing often also a wide range of adverse effects to the patients with already weakened health status [2,3,4]. Therefore, novel, safe and more efficient treatment modalities are highly needed and must be developed, whereas one possibility for this is to focus more on the natural plant-derived agents [5].

In ethnomedicine, natural products have been used for the management of both benign as well as malignant neoplasms already for centuries [6,7,8,9]. This approach has led to launching the screening program of plant-derived compounds in the US National Cancer Institute in the middle of the 20th century, resulting in the isolation and development of several anticancer drugs such as vincristine, vinblastine, and paclitaxel among others, currently approved for clinical use against different cancer types [10,11,12,13]. This success clearly demonstrates that nature is an important resource for anticancer agents, encouraging to continue such efforts.

Cucurbitacins (designated by the letters A, B, C, D, E, F, G, H, I, J, K, L, O, P, Q, R, S) constitute a class of natural triterpenoids [14]. These compounds can be found in many plants from the genera *Bryonia*, *Cucurbita*, *Cucumis*, *Echinocystis*, *Luffa*, *Citrullus* and *Lagenaria*, conferring a bitter taste in cucumber [14,15]. Several recent preclinical studies have demonstrated that this group of phytochemicals can exert antitumor activities in a variety of experimental models of different malignancies, including lung cancer [16], gastric cancer [17], colorectal cancer [18], liver cancer [19], pancreatic cancer [20], ovarian cancer [21], cervical cancer [22] and melanoma [23]. Such anticancer effects are achieved via interaction of cucurbitacins with multiple molecular targets and intervening in diverse cellular signaling cascades, suggesting that a high potential of this compound should be developed as a therapeutic tool.

To appreciate and draw more attention to these ancient molecules, this review article is focused on the different anticancer activities of cucurbitacins in diverse experimental model systems, describing anti-inflammatory, cell-cycle arresting, proapoptotic, antiangiogenic and antimetastatic effects of these triterpenoids. Moreover, co-effects of cucurbitacins with conventional cancer drugs are considered, presenting the most potent combinations for further studies. In addition, the possibilities to overcome the low bioavailability issues characteristic for natural compounds by modern nanotechnological methods are also discussed. In this way, the present review provides a strong basis for moving on with in vivo animal studies and human clinical trials, hopefully to apply cucurbitacins in the clinical settings in the future.

## 2. Chemistry of Cucurbitacins

Plants of the *Cucurbitaceae* family produce a class of biological substances that are known as cucurbitacins. Cucurbitacins are created by these plants to protect them from herbivores. Cucurbitacin A, B, C, D, E, F, I, L, 23, 24 dihydrocucurbitacin F, and hexanorcucurbitacin F, as well as the three acetylated derivatives, are the naturally occurring cucurbitacins [24]. Cucurbitacins are tetracyclic terpenes with steroidal structures in their chemical configuration. Cucurbitacins’ basic chemical structure is cucurbit-5-ene with a ring skeleton of 19(109)-abeo-10-lanost-5-ene (Figure 1). For instance, in the structure of cucurbitacin I (9,10,14-trimethyl-4,9-cyclo-9,10-secocholesta-2,5,23-triene) triene are replaced by hydroxy groups at positions 2, 16, 20 and 25 and oxo groups at positions 1, 11 and 22. This distinguishes the cucurbitacins from most other tetracyclic triterpenes. In all chemical structures of cucurbitacins, the presence of a 5,(6)-double bond is observed as a common characteristic. Cucurbitacins differ from steroidal nuclei in that their methyl group is located at carbon 9 instead of carbon 10 [14].

## 3. Absorption and Metabolism of Cucurbitacins

The presence of cucurbitacins is observed mainly in plant roots and fruits. They are a highly diverse group and divided into 12 categories, cucurbitacins A–T [25]. Less research has been conducted on cucurbitacins’ absorption, distribution, metabolism, and excretion; this is a topic that should be investigated considering the compound’s potential toxicity to mammals. [14]. Oral administration of cucurbitacin B (CuB) has been studied and shown to be slowly absorbed and metabolized in the gut [26]. CuE at a concentration of 100–200 μg/kg treatment had a plasma half-life of about 58–72% [27]. Similarly, oral bioavailability of CuB has been studied to be ~10% with plasma concentration ranging from 4.85 to 7.81 μg/L after 30 mins of oral dosing. After intravenous administration, it is distributed in large volume ~51.65 l/kg and exhibits a high tissue to plasma concentration ratios of ~60–280-folds in many organs. A negligible amount (~1%) of CuB was detected in urine and feces, and it was suggested that it probably undergoes biotransformation prior to excretion [28]. Studies have shown that they reach highest plasma concentration within 1.75 h and an elimination half-life of ~2.5 h. As oral delivery and absorption of cucurbitacins remain a great challenge, recent focus has been on the use of nano-micelles co-modified with cucurbitacins, which enhances the relative bioavailability of CuB by ~2.14–3.43 times [29]. However, further pharmacokinetic studies comprising metabolism and distribution of cucurbitacins are still required.

## 4. Anti-Cancer Mechanisms of Cucurbitacins

### 4.1. Apoptotic and Cell-Cycle Arrest

Apoptotic cell death can be triggered in cancer through internal and extrinsic processes, which converge on the control of caspase-dependent proteolysis of cellular proteins and DNA fragmentation [30,31,32]. Similarly, all tumor types have abnormal cell-cycle progression activity, which acts as a catalyst for carcinogenesis [33]. Recent research has shown that a variety of biological processes are regulated by cell-cycle proteins [34,35,36]. Therefore, numerous chemo-preventive FDA-drugs have been shown to mediate antitumor effects either via activation of apoptotic or cell-cycle arrest (Figure 2) signaling pathways [37,38,39]. For instance, results from Li et al. (2018) revealed that cucurbitacin I caused lung cancer (A549) cells to undergo excessive ERS, CHOP-Bax and caspase-12-dependent ERS-associated apoptosis [40]. In colorectal cancer (SW480 and Caco-2) cells, treatment with cucurbitacin B resulted in cell-cycle arrest at the G1 phase as well as decreased Cyclin D1 and Cyclin E1 levels. Both CRC cell lines underwent in vitro cell death when exposed to CuB, which was accompanied by caspase-3 and cleaved PARP [41]. Using triple negative breast cancer (TNBC), cucurbitacin E strongly boosted JNK activation while considerably decreasing AKT and ERK activation in MDA-MB-468 cells. It also significantly decreased expression of Cyclin D1, Survivin, XIAP, Bcl2 and Mcl-1 [42]. In the pancreatic cancer cell line Capan-1, CuD induced cell-cycle arrest and death via the ROS/p38 pathway [43]. Cucurbitacin I-induced cell death in ovarian cancer (SKOV3) included apoptosis, as evidenced by upregulated caspase 3 and BAX and a decrease in Bcl2 [21]. Flow cytometric measurement of DNA content and RT-PCR analyses suggested that cucurbitacin B caused G2/M arrest in human breast cancer cell lines (MDA-MB-231 and MCF-7) through elevated p21 expression [44]. Huang et al. showed that in human bladder cancer (T24) cells, cucurbitacin E-induced G2/M arrest was accompanied by a significant rise in p53 and p21 levels and a fall in the levels of STAT3, cyclin-dependent kinase 1 (CDK1) and cyclin B [45]. In addition, cucurbitacin E-induced G2/M phase arrest and death of T24 cells also depended on Fas/CD95 and mitochondria-dependent apoptotic pathways. Similarly, using other cancerous cell lines, cucurbitacins target the cell-cycle actions that involves growth inhibition, cell-cycle arrest at G2/M phase and induction of apoptosis [46]. Cucurbitacin I has been observed to suppress phosphotyrosine STAT3 in human cancerous lung cells [47]. Recently, it was observed to promote gastric cancer cell apoptosis by inducing the production of cellular ROS, as well as the endoplasmic reticulum stress pathway [40,48]. While cucurbitacin B, E and I have been observed to inhibit both JAK2 and STAT3 activation, cucurbitacin A and I have been reported to inhibit JAK2 and STAT3, respectively [47]. Treating Hep-2 cells with different concentrations of cucurbitacin B for various time intervals showed reduction in cell proliferation, cell-cycle distribution, and increased cell apoptosis in cancerous cell lines [46]. This study also stated that cucurbitacin B exhibited significant efficacy in inhibiting cell growth, arresting cell cycle at the G2/M phase, and inducing apoptosis in a dose- and time-dependent manner [46]. Similarly, cucurbitacins B, D, E were observed to inhibit proteins such as JAK-STAT3. They also inhibited mitogen-activated protein kinases (MAPK)- signaling pathways and tumor angiogenesis [48]. A study conducted on human umbilical vascular endothelial cell lines revealed cucurbitacin to significantly inhibit the proliferation, migration, and angiogenesis. It also blocked essential proteins such as Jak2-signal transducer, vascular endothelial growth factor receptor (VEGFR) and STAT3 signaling pathways [49]. Such studies have highlighted that the main mechanism involved in imparting the anti-tumorigenic potentials of cucurbitacins involves inhibition of the JAK/STAT3 signaling pathway, which plays an essential role in activation, proliferation, and maintenance of cancerous cells [14]. Another recent study has shown that treatment with 8 µM cucurbitacin IIb for 24 h remarkably inhibited the proliferation of HeLa and A549 tumor cells, with IC50 values of 7.3 and 7.8 µM, respectively, while increasing total apoptosis by 56.9 and 52.3%, respectively [50]. Another pathway by which cucurbitacin IIb induces apoptosis and cell-cycle arrest is by the regulating EGFR/MAPK pathway [51]. Similarly, cucurbitacin D was observed to regulate the levels of oncogenic signaling cascades, JAK/STAT, Wnt/β-catenin and associated non-coding RNAs in many cancer cell lines [52]. Recent studies have shown that CuIIb and cucurbitacin B induced apoptosis in cervical cancer cell lines by Nrf2 inhibition, whereas in lung cancer cell lines cucurbitacin B was responsible for suppressing growth and inducing apoptotic death by impeding IL-6/STAT3 signaling [21,53].

### 4.2. Antiangiogenic and Antimetastatic Mechanisms

The physiological process by which new blood vessels develop from pre-existing vessels is known as angiogenesis. Anti-angiogenesis causes suppression of tumor growth because of hunger and toxic waste buildup in its microenvironment [54,55]. The development and metastasis of the tumor have a major impact on the cancer vasculature (Figure 3). Vascular endothelial growth factors (VEGFs) are crucial protein regulators of angiogenesis and metastasis. Studies have shown that inhibiting the VEGFR2-mediated JAK/STAT3 pathway is considered as an effective approach to suppress angiogenesis [49]. Though many studies about the mechanism of cucurbitacins and angiogenesis are not well known, few studies have still shown that cucurbitacins such as cucurbitacin B, cucurbitacin D, cucurbitacin E and cucurbitacin I possess anti-angiogenesis properties [56,57]. CuB significantly inhibited angiogenesis, metastasis, and vascular development in dose-dependent manner in in vivo models and chick embryos [56]. CuE significantly inhibited human umbilical vascular endothelial cell (HUVEC) proliferation and angiogenesis by targeting the VEGFR2-mediated Jak2/STAT3 signaling pathway [49]. CuB has been observed to inhibit ERK1/2, prevent Raf-MEK-ERK from activating STAT3, which ultimately plays a key role in angiogenesis [58]. A similar effect of CuB was seen also in human breast cancer cell lines. It successfully inhibited angiogenesis by targeting the FAK/MMP-9 signaling axis [59]. CuB showed antimetastatic activity and targeted angiogenesis also in paclitaxel resistant A2780/Taxol ovarian cancer cells. It also suppressed angiogenesis by downregulating the expression of HIF-1 targets, VEGF, VEGFR2 phosphorylation and erythropoietin [56,60]. Another study revealed the effective use of CuE for anti-angiogenesis in Huh7 cells. It decreased the tube formation in HUVECs and was also responsible for inhibiting the process of neo-vascularization in CAM assays [61]. A recent study showed that CuE modulated the JAK/STAT3 pathways, which regulated the angiogenesis [62]. CuE has been also involved in inhibiting the KDR/VEGFR2-mediated pathway of angiogenesis [63]. Treating A549 cells with cucurbitacins for ~21 days showed positive results for inhibiting metastasis by regulating the levels of cyclooxygenase 2, matrix metalloproteinase 9, and cyclin D11 [64]. Similarly, other cucurbitacins were observed to inhibit angiogenesis in MDA-MB-231 and MCF-7 cancer cells by inhibiting the JAK/STAT pathways [65].

### 4.3. Anti-Inflammatory Mechanisms

Most malignancies’ growth and malignant progression are correlated with inflammation [66,67,68]. Both intrinsic and extrinsic inflammations have the potential to inhibit the immune system, which creates an ideal environment for the growth of tumors [69,70,71]. As a result, focusing on inflammation is a tempting strategy for both cancer therapy and cancer prevention [69,72]. Cucurbitacins have been observed to interact with proteins associated with inflammatory (Figure 4) pathways such as interleukins (IL)-6, IL-5, IL-1β, IL-12, IL-13 in a dose-dependent manner [73]. Dietary cucurbitacin E has been shown to reduce inflammation and immunosuppression by downregulating the NF-κB signaling pathway [74]. CuB has been studied to possess protective effects by reducing inflammatory responses on sepsis-induced acute lung injury in in vivo rat models. It significantly reduced the levels of TNF-α, IL-6, cytokine secretion and accumulation of inflammatory cells. It also regulated the levels of Ca^2+^, which play an essential role in inflammatory responses [75]. CuB inhibited inflammatory responses through targeting the SIRT1/IGFBPrP1/TGF β1 axis. It downregulated the expression levels of TGF β1, IGFBPrP1, and upregulated the expression of SIRT 1 [76]. Similarly, CuE decreased the levels of pro-inflammatory cytokines, such as IL-17 and IFN-γ, as well as the activities of the STAT3 and IL-17A-promoter in allo-reactive T cells [77]. CuE has been shown to inhibit skin inflammation and fibrosis by regulating the expression of α-Sma and Col-I in mice models [76]. Recently, it has also been demonstrated that CuE ameliorated lipopolysaccharide-evoked injuries and inflammation in bronchial epithelial cells by regulating the TLR4-NF-κB signaling. It was responsible for suppressing levels of inflammatory cytokine production, TNF-α, IL-6 and IL-8 [78]. Cucurbitacin B was observed to directly bind to toll-like receptor 4 (TLR4) and activate NLRP3 inflammasome, which further ultimately executed pyroptosis in A549 cells. CuB treatment has been observed to upregulate the protein expressions of IL-1β, GSDMD, HMGB1 and led to inhibition of generation of mitochondrial ROS and pyroptosis [79]. CuB was reported to sensitize CD133+ HepG2 cells in in vitro and in vivo models [80].

## 5. Synergistic Effects with Other Drugs

During cancer therapies, the side effects and long-term consequences of anti-cancer chemotherapy continue to be a major cause of concern. The effectiveness of current medications to prevent the negative effects of chemotherapy is frequently insufficient. As a result, the current cancer treatment pattern is shifting toward combination chemotherapy. A combination of medications improves the possibility that numerous oncogenic and resistance signalings will be inhibited simultaneously, whereas chemotherapeutics can affect cancer cells by affecting only one or two stages in the cell cycle. A modified cell line and the development of drug resistance are less likely when the malignant cells are attacked via several chemo-preventive agents. Recent research using the ovarian sarcoma M5076 cell line demonstrated that the synergism of cucurbitacin I and doxorubicin enhanced cytotoxicity and reduced the volume and weight of tumor cells. It was observed that treatment with CuI and doxorubicin decreased glutathione (GSH) levels, enhancing cytotoxicity in tumors. The phytochemical also increased DOX-induced antitumor activity [81]. In both in vitro and in vivo investigations, the combination of cucurbitacin B and curcumin proved particularly efficient against hepatocellular cancer. It encouraged apoptosis and reduced the potential for multidrug resistance in human hepatocarcinoma cells. Significant activity was observed with 2:1 ratio (cucurbitacin B:curcumin). It led to changes in tumor volume, caspase3 activation and ATP down-regulation, thereby serving as a novel, promising approach for treating human hepatoma [82]. Irinotecan and cucurbitacin have been found to have synergistic effects on the ability of colon cancer cell lines to resist proliferating, which together have increased their therapeutic benefits by activation of JAK2/STAT3, which plays a crucial role in cell survival and proliferation [83]. The synergistic effect of 23, 24-dihydrocucurbitacin B and cucurbitacin R was observed on inhibiting the expression of TNF-α, IL-6 through the NF-κB pathway in HepG2 cell line. Similar effects were observed with cucurbitacin D and docetaxel, which together effectively inhibited cancer cell growth and the cloning potential of prostate cancer stem cells. This combination has been suggested to be a novel therapeutic modality for the treatment of advanced prostate cancer [84,85]. Cucurbitacin B along with gemcitabine has been observed to induce apoptosis of MDA-MB-231 breast cancerous cells by regulating JAK/STAT3, Bcl-Xl, cyclin A and B1 [86], while with imatinib-mesylate it was responsible for inhibiting the proliferation of cells and inducing apoptosis through inhibition of MMP-2 expression [87]. Administration of cucurbitacin B and higenamine (in ratios 1:1, 1:2 and 2:1) significantly increased the cytotoxic effects on breast cancer cell lines, which also increased apoptosis and cell-cycle arrest in G2/M. This combination acts on essential proteins such as Akt farnesyl-transferase, platelet-derived growth factors, cyclin A2, CDK2, etc [88]. Cucurbitacin E with doxorubicin successfully induced apoptosis, cell-cycle arrest and autophagy [89]. Recent advances in the study of cucurbitacin IIb (CuIIb) and kinoin A (KinA) from *Ibervillea sonorae* (S.Watson) Greene highlight the ability of these molecules to reduce proliferation and to tempt apoptotic and cell-cycle apprehension in tumors; also, the levels of STAT3 expression were downregulated after treatment with CuIIb. These findings imply that CuIIb and KinA may be considered in future research for the creation of efficient and secure anti-cancer treatments for breast, cervical, gastric and other cancer types where STAT3 is overexpressed [90].

## 6. Safety Studies

As several researchers have identified the presence of toxic cucurbitacins in the roots, leaves, and fruits of some plants, such as *Combretum zeyheri* Sond and *Cucumis anguria* L. [91], further studies are highly needed to evaluate the safety issue and determine the value of the no-observed-adverse-effect-level (NOAEL), before any pharmacological applications of cucurbitacins as anticancer agents can be recommended. On the other hand, it has been claimed that safety issues of cucurbitacins may be related to their specific variants, the purity of preparations and differences in the study models [92]. For example, no toxic reactions were observed in lung cancer xenografted mice treated intraperitoneally with 1 mg/kg of cucurbitacin B [93] or cucurbitacin Q [94]. Today, it is generally accepted that the active dose and lethal dose are not the same for different types of cucurbitacins, being related to their structural peculiarities [92]. Therefore, further clinical trials on the safety and efficacy of specific variants of cucurbitacins are highly required. Table 1 and Table 2 present an outline of diverse antiproliferative actions of cucurbitacins.

## 7. Conclusions and Further Perspectives

As demonstrated in the present review article, diverse variants of cucurbitacins can be active against different types of malignancies. Moreover, the combination of cucurbitacins with conventional chemotherapeutic drugs might lead to synergistic anticancer effects, revealing a great promise for the application of these compounds in future clinical settings as either individual agents or drug adjuvants. However, many steps have remained to achieve this attractive goal. First, the safety of cucurbitacins must be elucidated, jointly with their metabolic conversion and possible bioactivities of various metabolites. Secondly, the proper dosage regimens also need to be elaborated. Studies related to the pharmacokinetic properties of cucurbitacins, including t1/2, Cmax, Tmax, Vd, mean residence time, etc., should be carried out singly or in synergism. Investigations on the metabolic conversions of cucurbitacins will improve its bioavailability and stability prospective. In this way, the current review presents a strong basis to move on with these next steps in the path to ultimately find novel, safe and more efficient therapies against cancer.

## Figures and Tables

**Figure 1 biomolecules-13-00057-f001:**
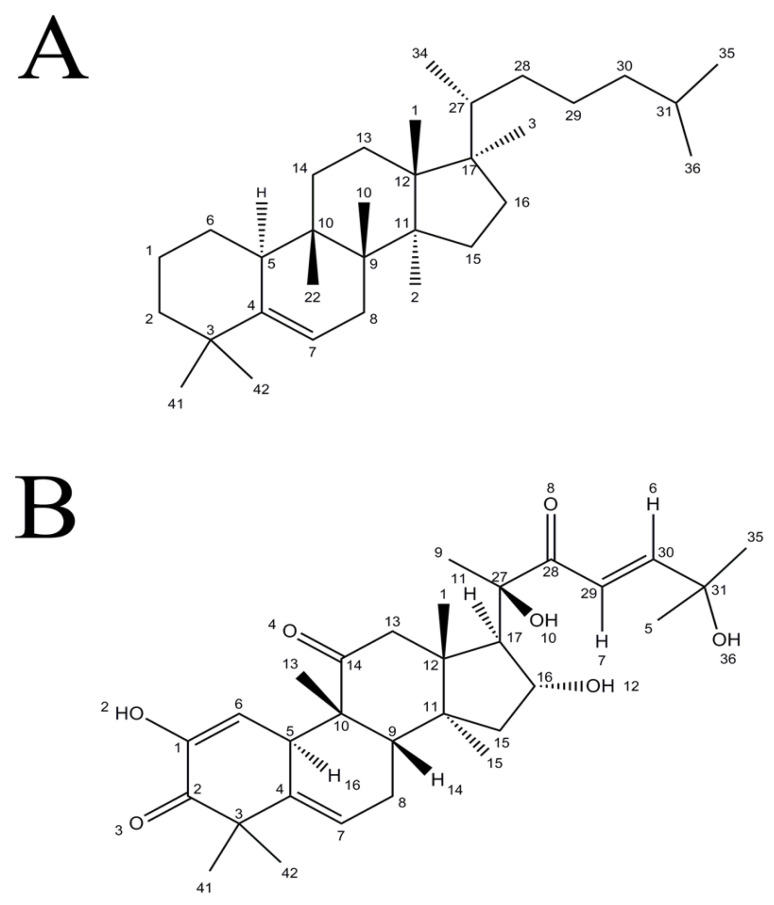
Basic skeleton of cucurbitacins. (**A**) (8R,9R,10S,13R,14R,17R)-4,4,8,9,10,13,14,17-octamethyl-17-((R)-6-methylheptan-2-yl)-2,3,4,7,8,9,10,11,12,13,14,15,16,17-tetradecahydro-1H-cyclopenta[a]phenanthrene (Cucurbitacin). (**B**) (8S,9R,10R, 13R,14S, 16R, 17R)-17-((R,E)-2,6-dihydroxy-6-methyl-3-oxohept-4-en-2-yl)-2,16-dihydroxy-4,4,9,13,14-pentamethyl-7,8,9,10,12,13,14,15,16,17-decahydro-3H-cyclopenta[a]phenanthrene-3,11(4H)-dione (Cucurbitacin I).

**Figure 2 biomolecules-13-00057-f002:**
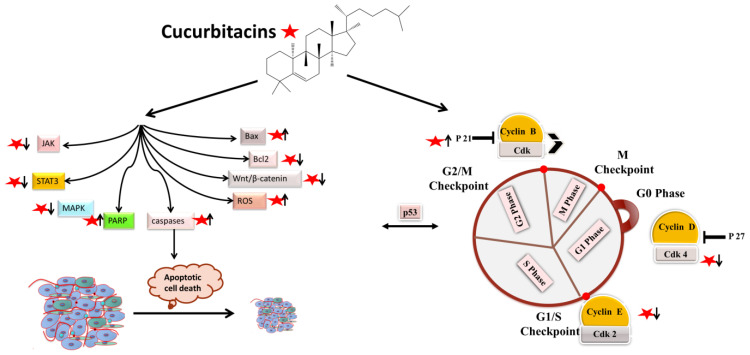
Molecular targets of cucurbitacins in modulating cell-cycle progression and inducing apoptotic cell death.

**Figure 3 biomolecules-13-00057-f003:**
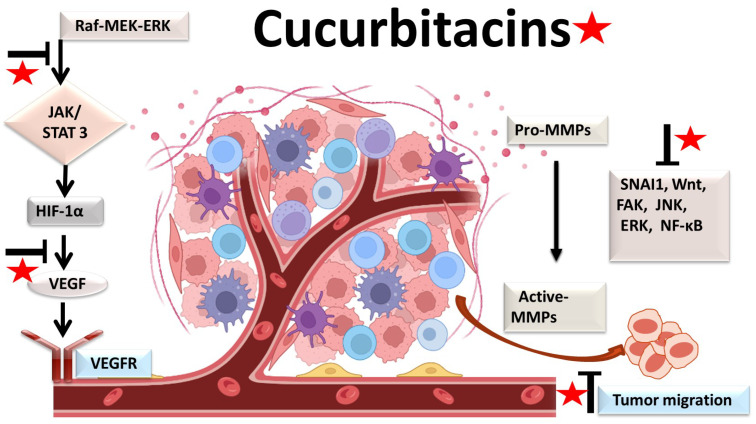
Molecular targets of cucurbitacins in suppressing angiogenesis and impeding cancer cell metastasis.

**Figure 4 biomolecules-13-00057-f004:**
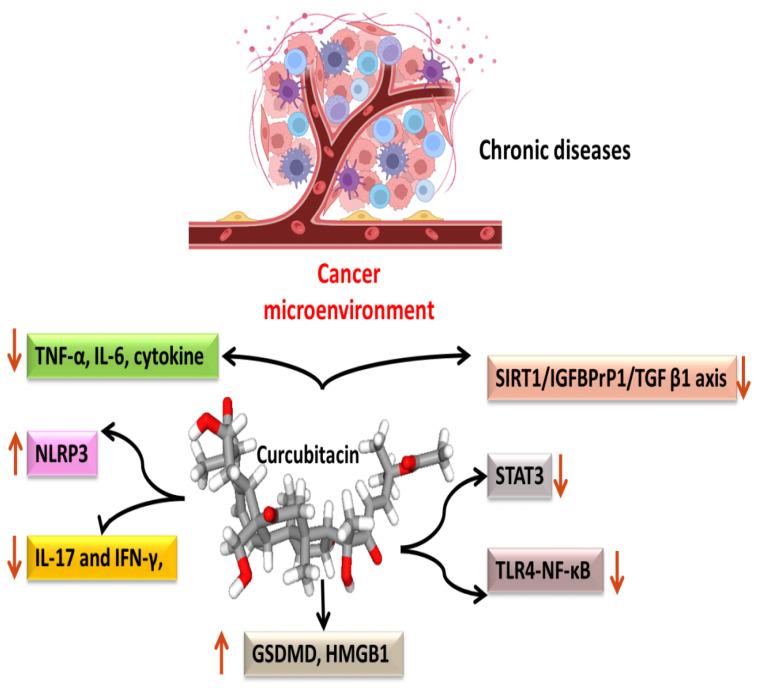
Anti-inflammatory targets of cucurbitacins in malignant cells.

**Table 1 biomolecules-13-00057-t001:** Antiproliferative actions of cucurbitacins using in vitro investigations.

Type of Tumor	Cell Lines	Effects	Mechanisms	Concentration	References
Neuroblastoma	SH-SY5Y	Regulation of cell cycle and induces apoptosis	↑ cell-cycle arrest at the G2/M phase, ↓ p-JAK2, ↓ p-STAT3, ↓ phospho-extracellular signal-regulated kinases, ↓ c-Jun N-terminal kinase, ↓ p38, ↓ MAPK, ↓ Cyclin B1 ↓ Bcl2- x, ↑ p53 and p21, ↓ Bcl-2 ↑ Bax	0–128 µM	[95]
Osteosarcoma	U-2 OS	Cell-cycle apprehension, apoptosis and inhibition of angiogenesis	↓ cell viability, proliferation, migration ability, ↓ MMP-2 and 9, ↑ apoptotic pathway, ↓ MAPK signaling andJAK2/STAT3 cascade, ↓ VEG F, ↑ caspase-3, -8 and -9, ↑ Bad and Bax, ↓ Bcl-2 and Bcl-xL, ↓ p38, ERK1/2, JNK and p-JNK	20, 40, 80 and 100 µM	[96]
Cholangiocarcinoma	KKU-452	Inhibits metastatic behavior	↓ FAK activation, ↓ phospho-FAK protein, ↓ migration, invasion and adhesion abilities, ↓ MMP-9, ICAM-1 and VEGF,	0, 5, 10, 25, 50, 100 nM	[97]
Tongue squamous	CAL27 and SCC9	Induced apoptosis and microRNA mediated	↓ proliferation, migration and invasion, ↓ Xinactive specific transcript (XIST), ↑ miR-29b	0.001, 0.01, 0.1, 1, 10, 100 µM	[98]
Laryngeal squamous	Hep-2	Regulation of cell cycle and apoptosis	compared with single treatment, combination treatment ↓ cell proliferation and viability, G2/M enrichment was accompanied by a reduction in G0/G1 phase cells, ↑ condensation of chromatin, ↑ nuclear fragmentations and apoptotic bodies, ↓ p-STAT3, ↓ Bcl-2, ↓ cyclin B1	1 µM cucurbitacin B + 2, 5, 10, 20, and 30 µM cisplatin	[99]
	Hep-2	Regulation of cell cycle, apoptosis, enhance docetaxel chemosensitivity	compared with single treatment, combination treatment ↓ cell proliferation and viability, G2/M enrichment was accompanied by a reduction in G0/G1 phase cells, ↑ condensation of chromatin, ↑ nuclear fragmentations and apoptotic bodies, significant activation (phosphorylation) of ERK1/2, ↓ p-STAT3, ↓ Bcl-2, ↓ cyclin B1	1 µM cucurbitacin B + 25 nM docetaxel	[100]
Colon	HCT116;	Apoptosis induction	↓ proliferation and migration ability, G2/M arrest, ↓ cyclin A, ↓ cyclin D1, ↑ p21, ↑ early apoptosis, ↓ p- Akt (Ser473)	0.001 μM–10 μM	[14]
Breast	MDA-MB-231	Suppresses metastasis	↓ migration, invasion and adhesion ability, ↓ p-FAK (focal adhesion kinase), ↓ p- paxillin, ↑ intracellular ROS generation,	0–100 nmol·L^−1^	[101]
	MCF-7	Induces autophagy	↓ cell viability, ↑ γH2AX, comet tails were significantly longer, ↑ phosphorylation of ATM (Ser-1981) and ATR (Ser428, ↑ LC3 II, ↓ p-mTOR, ↓ p-Akt (Ser308 and Ser473), ↓ p62, ↑ Beclin-1 and p-ULK1 (Ser 317), ↑ intracellular ROS	0–200 nM	[102]
	MCF-7	Regulation of cell cycle and induces apoptosis	↑ γH2AX, comet tails were significantly longer, ↑ phosphorylation of ATM (Ser-1981) ↑ p- p53(Ser-15)	0–800 nM	[103]
Pancreatic	ASPC-1,BXPC-3, CFPAC-1, SW 1990	Induced cell-cycle Arrest and apoptosis	↓ proliferation, viability, ↑ percentage of cells in G2/M phase ↓ decrease in S and G0/G phase cells, ↓ cyclin B1, ↓ cyclin D1 and cyclin A2, ↓ Caspase3 and PARP1, ↓ p-JAK2 andp-STAT3	0, 0.25, 0.5 and 1.0 μM	[19]
	AsPC-1, BxPC-3, CaPan-1, and HPAF-II	Cell-cycle arrest	↓ viability of PanCa cells, ↓ colony formation capacity, ↑ G2/M Phase, ↓ invasion and migration ability, ↓ MUC13, restores miR-145 expression, ↓ proliferation of gemcitabine resistant PanCa cells, ↓ RRM1/2 expression,	0.1, 0.25, and 0.5 µM	[104]
Hepatoblastoma	HepG2	Induced cell-cycle arrest and apoptosis	↓ proliferation and migration ability, G2/M arrest, ↓ cyclin A, ↓ cyclin D1, ↑ p21, ↑ early apoptosis, ↓ p- Akt (Ser473), ↑ caspase-8 and PARP,	0.001 μM–10 μM	[14]
Lung	A549 cells	Inhibits Metastatic Behavior, Cell-cycle arrest and apoptosis	↓ proliferation and cell viabilitygradually, ↑ condensation, ↓ p-STAT3 levels, ↑ percentage of cells in G2/M phase, ↑ levels of ROS, ↑ membrane pore formation, ↑ pyroptosis, ↑ caspase-3, 9 activity	Cucumber-derived nanovesicles (CDNVs)- CDNVs containing 10 nM CuB	[15]
	A549	Apoptosis and microRNA mediated	↓ proliferation of lung cancer cells, ↑ cell apoptosis frequency, ↑ Bax and cleaved caspase3, ↓ cyclin B1 and Bcl-2, ↓ XIST and IL-6, ↑ miR-let-7c expression, ↓ IL-6/STAT3 pathway	0.1, 0.3, 0.6, and 0.9 μM	[52]
	A549, A549-GR	Anti-metastasis	↓ EMT, ↓ p-PI3K, ↓ p-Akt, ↓ p-mTOR	5, 10, 15, 20 nM	[105]
	PC9 (gefitinib resistance)	Apoptosis and microRNA mediated	↑ miR-17-5p in PC9/GR cells, caspases, STAT3, ↓ p- STAT3	--	[106]
	A549	Induced cell-cycle arrest and apoptosis	↓ proliferation and migration ability, G2/M arrest, ↓ cyclin A,↓ cyclin D1, ↑ p21, ↑ early apoptosis, ↓ p- Akt (Ser473)	0.001 μM–10 μM	[14]
	A549, NCI-H1299 (H1299), NCI-H1975 (H1975), and NCI-H820 (H820) (gefitinib resistance)	Induces apoptosis	↓ anchorage-dependent growth and clonogenic ability, ↓ proliferation, invasion and migration ability, ↑ caspase-8 and 3, caspase-3, c-PARP, ↑ Lysosomal Degradation of EGFR and thus Inhibits ERK Signaling, ↓ CIP2A expression, ↑ PP2A, ↓ pAkt	IC50 (µM) H1299-0.77 A549-0.76 H197-0.63 H820-0.19	[107]
	A549	Induces apoptosis	↓ cell proliferation, ↑ capase-3/9, ↓ PI3K, p-AKT and p-p70S6K	0, 50, 100 and 200 nM	[108]
	A549	Induces autophagy	↓ cell viability, colony formation ability, ↑apoptosis frequency, ↑ apoptotic and necrotic, ↑ autophagosomes, ↑ LC3-II/LC3-I, ↓ p-mTOR (S2448), ↓ p- ERK and p- STAT3	0, 100, 200, 300, 400 and 500 nM	[109]
	A-549	Induces cell-cycle arrest and apoptosis	↑ G2/M phase cell-cycle collapse, ↓ m-TOR/PI3K/Akt proteins	0, 10, 20, 40, 100, 150 and 200 μM	[110]
	A549	Regulation of cell cycle and induces apoptosis	↓ proliferation and colony forming ability, ↑ γH2AX, comet tails were significantly longer, ↑ G2/M phase, ↑ phosphorylation of ATM (Ser-1981), ↑ p- p53(Ser-15)	0–800 nM	[103]
	A549	Regulation of cell cycle and induces apoptosis	↓ cell proliferation, ↑ G2/M phasecells, % early and lateapoptotic cells, ↑ cell shrinking, ↑ intracytoplasmic vacuoles, ↑chromatin condensation, ↑mitochondrial swelling, ↑caspase-3 and caspase-9, ↑ disruption of the ΔΨm, ↓ (p)-STAT3, ↓ cyclinB1 ↓ Bcl-2	0.02,0.1, 0.5, 2.5, 12.5 and 62.5 µmol/L	[111]
	CD133-positive and CD133-negative	Inhibits tumorigenic ability and enhances radiochemo-sensitivity	↓ STAT3, ↓ tumorigenic capacity,↓ sphere formation ability, ↓ radioresistance and chemoresistance in CD133-positive, ↓ stemness gene signature of CD133-positive, ↓ Bcl-2, ↓ Bcl-xL ↓survivin, ↑ Bax	radiation doses (0 Gy, 2Gy, 4 Gy, 6 Gy, 8 Gy, and 10 Gy) + 0, 50, 100 and 150 nM	[112]
Gastric	Human DDP-resistant gastric cancer cell lineSGC7901/DDP and human GC cell line SGC7901	Induces autophagy and apoptosis	↓ viability, ↓ clonogenic ability, ↑ cytoplasmic shrinkage, ↓ pro-caspases-3 and -9 and cleaved PARP, ↑ LC3 II and Beclin1, ↓ P-gp, and HIF-1α, ↓ phosphorylation of mTORC1 effectors (mTOR, p70S6K and 4E-BP1), ↓ pAkt, ↑PP2A, ↓ CIP2A	IC50 (nM) SGC7901- 216.70 SGC7901/DDP- 170.25	[113]
	MKN-45	Cell-cycle arrest, apoptosis	↓ proliferation of cancer cells, ↑ progression of the cell cycle from G0/G1 to S phase, ↓ cyclin D1, ↓ cyclin E, ↓ CDK4 and CDK2 (cyclin-dependent kinase), ↑ p27, ↑ cell apoptosis frequency, ↑Bax, ↓ Bcl-2, ↓ JAK2/STAT3 signaling pathway	0.1, 1 or 10 µM	[114]
Colorectal	SW620 and HT29	Regulation of cell cycle and apoptosis	↑ proliferation and invasion, ↑ chromatin condensation and fragmentation, ↓ pro-cas-3, ↓cleaved PARP, ↓ YAP ↓Cyr 61 and c-Myc, ↑ LATS1	0, 0.1, 0.2, 0.4, 0.6, 0.8 and 1.0 μM	[115]
Prostate	LNCaP, DU145, and PC-3;	Induced cell-cycle arrest and apoptosis	↓ proliferation and migration ability, G2/M arrest, ↓ cyclin A,↓ cyclin D1, ↑ p21, ↑ early apoptosis, ↓ p- Akt (Ser473), ↑ caspase-8 and PARP	0.001 μM–10 μM	[15]
Bladder	MB49	Induce the cell death pathways, apoptosis and autophagy	↓ viability of cancer cells, ↓Bcl-2, ↑ LC3II, ↓ phosphorylation of p27, PRAS40 and Raf-1 proteins, ↑ p- AKT, ↓ p-ERK1/ ERK2, ↓ p-mTOR, ↑ BAD, ↑AMPKα	CuB- 0.01–50 μM and Cisplatin- 0.5–50 μM	[116]
	T24	Induced cell-cycle arrest and apoptosis	↓ proliferation and migration ability, G2/M arrest, ↓ cyclin A,↓ cyclin D1, ↑ p21, ↑ early apoptosis, ↓ p- Akt (Ser473), ↑ caspase-8 and PARP	0.001 μM–10 μM	[14]
Ovarian	SKOV3	Cell-cycle arrest, apoptosis	↑ chromatin condensation, ↑ apoptotic body formation, ↑ deformed cell morphology, ↑ intracellular ROS levels, ↓ MMP, ↓ mTOR/PI3K/Akt signaling pathway, ↓ m-TOR, ↓ phospho m-TOR proteins. ↓ PI3K/Akt protein expressions, ↑ DNA damage, ↑ cell-cycle arrest at G2/M checkpoint	0, 10, 20, 40, 80 and 160 μM	[117]

**Table 2 biomolecules-13-00057-t002:** Antiproliferative actions of cucurbitacins using in vivo investigations.

Type of Tumor	Animal Models	Effects	Mechanisms	Dosage	Duration	References
Laryngeal	Nude mice injected with 5 × 10^6^ Hep-2 cells	Inhibits tumor growth	↓ tumor weight and volume,	55 μg/kg/day ofcucurbitacin B, or 7.5 mg/kg/week of docetaxel	14 days	[100]
	Athymic nude mice 5 × 10^6^ Hep-2 cells	Inhibits tumor growth	↓ tumor growth inhibition, no significant adverse effects were observed	55 µg/kg daily for 14 days, orcisplatin (intraperitoneal injection) 10 mg/kg daily	14 days	[99]
Tongue squamous	Nude mice injected with SCC9 cell lines (8 × 10^6^)	Inhibits tumor growth	↓ XIST, XIST expression was lost in XIST KO (knockout)	0.5 mg/kg	14 days	[98]
Lung	C57BL/6 J mice injected with 5 × 10^5^ B16-F10-Luc cells	Anti-metastasis	↓ lung index, ↓ lung metastasis	CuB (0.25 mg/kg, and 0.5 mg/kg) groups and Geftinib (40 mg/kg)	14 days	[105]
	BALB/c nude mice injected with A549 cells (2×10^6^ cells)	Inhibits tumor growth	↓tumor weight, ↑necrotic and apoptotic cells were observed in tumor sections, ↓ p-STAT3, ↓ CD31,	CsDNVscontaining 472 nM CuB	14 days	[15]
	nu/nu mice injected with GR NSCLC H1975 cells (2.5 × 10^6^)	Inhibits tumor growth	↓ tumor weight, ↓ CIP2Aand EGFR	gefitinib (30 mg/kg) or CucB 0.5 mg/kg	24 days	[107]
	BALB/c mice injected with 1 × 10^5^ CD133-positive and CD133-negative cells	Inhibited tumor growth	↓ lung metastasis, ↓ tumor size, ↓ tumorigenic and metastatic capabilities, ↑ survival rate	1 mg/kg	4 weeks	[112]
Pancreatic	BALB/c-nu injected with BXPC-3cells (2 × 10^6^ cells)	Inhibits tumor growth	↓ p-STAT3 in the tumors of mice, ↑ PCNA in control	1 mg/kg and 2 mg/kg	30 days	[19]
	NOD-SCID gamma mice injected with HPAF-II cells (4 × 10^6^ cells)	Inhibits tumor growth	↓ MUC13, ↓ PCNA (nuclear proliferating cell antigen), ↑ miRNA-145	1 mg/kg	40 days	[104]
Hepatocellular	SCID mice injected with HepG2 (3 × 10^6^)	Inhibits tumor growth	No significant differences of body weight, ↑ DNA cleavage, ↑ CDKN1A (p21), ↑ CDKN1B (p27), ↑ FOXO, ↑ p- Akt	0.1 mg/kg	3 weeks	[14]
Prostrate	SCID mice injected with HepG2 (3 × 10^6^) cells	Inhibits tumor growth	No significant differences of body weight, ↑ DNA cleavage, ↑ CDKN1A (p21), ↑ CDKN1B (p27), ↑ FOXO, ↓ p- Akt	0.1 mg/kg	8 weeks	[14]
Bladder	C57BL/6 injected with 1 × 10^6^ MB49 cells	Reduced the tumor growth	No histopathological changes, ↓ Bcl-2, ↑ LC3II, ↓ phosphorylation of p27, PRAS40 and Raf-1 proteins, ↑ p- AKT, ↓ p-ERK1/ ERK2, ↓ p-mTOR,	CuB (0.5–1 mg/kg) and Cis (1–3 mg/kg)	19 days	[116]

## Data Availability

Not applicable.

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
