# Peer review of "Cucurbitacins as Potent Chemo-Preventive Agents: Mechanistic Insight and Recent Trends"

_biomolecules, 2022, doi:10.3390/biom13010057_

Round 1

Reviewer 1 Report

In this review the authors focused on the different antineoplastic effects of cucurbitacin in a number of models. The paper is generally comprehensively described; however, the work need to be better organized:

-          Paragraph 3: the authors are supposed to write about absorpion and metabolism of curcubitacin, but informations about mechanisms of action or toxicity are also furnished and, in my opinion should be moved in the related paragraph

-          Paragraph 5: a number of interesting results concerning the synergistic effects of curcubitacin and other drugs/compounds are reported without any conclusion/comment about them. Only for the last study mentioned is a conclusion drawn. Maybe it is a conclusion of the cited work, however the authors shoud make comments/conclusion of every work they have cited or none at all.

-          Paragraph 6: Table 1 and 2 which show an outline of the various effects of curcubitacin shold be moved at the end of the part on the mechanisms of action, unless details on the presence or absence of toxic effects are also added.

-          Paragrapf 7: some conclusions on pharmacokynetics should be added

Author Response

Manuscript ID: biomolecules-2095947

Type of manuscript: Review

Title: Cucurbitacin as a potent chemo-preventive agent: Mechanistic insight

and recent trends

The authors of this manuscript are immensely thankful to the reviewers, managing editor and editor in chief, for reconsidering the manuscript in your esteemed journal “Biomolecules”. The authors feel that the comments and suggestions given by the reviewers have up-graded the level of the manuscript. Please find the enclosed herewith author’s response. All changes are highlighted in green color in the text.

Reviewer 1:

In this review the authors focused on the different antineoplastic effects of cucurbitacin in a number of models. The paper is generally comprehensively described; however, the work need

to be better organized:

Comment 1: Paragraph 3: the authors are supposed to write about absorpion and metabolism of curcubitacin, but informations about mechanisms of action or toxicity are also furnished and, in my opinion should be moved in the related paragraph

Response: We agree with reviewers therefore, the text furnished about mechanisms and toxicity have been removed. Line no. 114-117

Comment 2: Paragraph 5: a number of interesting results concerning the synergistic effects of curcubitacin and other drugs/compounds are reported without any conclusion/comment about them. Only for the last study mentioned is a conclusion drawn. Maybe it is a conclusion of the cited work, however the authors shoud make comments/conclusion of every work they have cited or none at all.

Response: We are thankful to the revierer for in depth observation. Therefore, the studies are further supplemented with conclusive remark. Line no. 266-269, 272-274, 276-277, 282-283, 289-291.

Comment 3: Paragraph 6: Table 1 and 2 which show an outline of the various effects of curcubitacin should be moved at the end of the part on the mechanisms of action, unless details on the presence or absence of toxic effects are also added.

Response: Authors agree therefore, these tables are recited in the mechanisms section. Line no. 251-252

Comment 4: Paragraph 7: some conclusions on pharmacokinetics should be added

Response: We further edited few lines in conclusions section to highlight the pharmacokinetics and future perspectives of cucurbitacin studies. Line no. 325-328.

Reviewer 2 Report

Comments:

1.       I suggest that authors number the provided structures to facilitate the reader. (Figure 1 A and B)

2.       The resolution of figures 2 and 3 could be improved.

3.       The authors should check the spelling, for instance, in the abstract, conclusion, etc.

4.       Checking Plagiarism should be done.

Author Response

Manuscript ID: biomolecules-2095947

Type of manuscript: Review

Title: Cucurbitacin as a potent chemo-preventive agent: Mechanistic insight

and recent trends

The authors of this manuscript are immensely thankful to the reviewers, managing editor and editor in chief, for reconsidering the manuscript in your esteemed journal “Biomolecules”. The authors feel that the comments and suggestions given by the reviewers have up-graded the level of the manuscript. Please find the enclosed herewith author’s response. All changes are highlighted in green color in the text.

Reviewers 2:

Comment 1: I suggest that authors number the provided structures to facilitate the reader. (Figure 1 A and B).

Response: We agree therefore, it has been mentioned. Line no. 102 & 105

Comment 2: The resolution of figures 2 and 3 could be improved.

Response: We agree with reviewer’s suggestions, therefore, quality of figure 2 and 3 further improved. Line no. 188 & 222

Comment 3: The authors should check the spelling, for instance, in the abstract, conclusion, etc.

Response: authors re edited the MS for spelling and grammar errors. All changes are highlighted in green color.

Comment 4: Checking Plagiarism should be done.

Response: It has been checked and within permissible range.

Reviewer 3 Report

In this manuscript, the authors aim to review the relevant studies of cucurbitacin on mechanisms of bioactivity. The authors have put many efforts and quite abundant information in this study. Many of the literatures review are promising and contributing the support of the conclusions. However, there are some critical concerns need to be addressed before accepted for publication.

1. In the beginning, the cited word of “cucurbitacin“ by the authors would cause misunderstanding to the composition as one component. It should be strengthened that the cucurbitacins are known and have been designated by the letters: A, B, C, D, E, F, G, H, I, J, K, L, O, P, Q, R and S (doi: 10.4103/0973-7847.156314). Therefore, the word of “cucurbitacins” was suggested to be cited in the manuscript if it was not specified to any compounds.

2. The occurrence of cucurbitacins in the natural resources should be discussed in more detail.

3. In the Figure 1, the authors present the basic skeleton of curcurbitacins, why the (B) CucurbitacinI is shown in the figure?  

Author Response

Manuscript ID: biomolecules-2095947

Type of manuscript: Review

Title: Cucurbitacin as a potent chemo-preventive agent: Mechanistic insight

and recent trends

The authors of this manuscript are immensely thankful to the reviewers, managing editor and editor in chief, for reconsidering the manuscript in your esteemed journal “Biomolecules”. The authors feel that the comments and suggestions given by the reviewers have up-graded the level of the manuscript. Please find the enclosed herewith author’s response. All changes are highlighted in green color in the text.

Reviewer 3:

In this manuscript, the authors aim to review the relevant studies of cucurbitacin on mechanisms

of bioactivity. The authors have put many efforts and quite abundant information in this study.

Many of the literatures review are promising and contributing the support of the conclusions.

However, there are some critical concerns need to be addressed before accepted for publication.

Comment 1: In the beginning, the cited word of “cucurbitacin“ by the authors would cause misunderstanding to the composition as one component. It should be strengthened that the cucurbitacins are known and have been designated by the letters: A, B, C, D, E, F, G, H, I, J, K, L, O, P, Q, R and S (doi: 10.4103/0973-7847.156314). Therefore, the word of “cucurbitacins” was suggested to be cited in the manuscript if it was not specified to any compounds.

Response: We agree with reviewer suggestions therefore, it has been updated throughout the text. All changes are highlighted in green color.

Comment 2: The occurrence of cucurbitacins in the natural resources should be discussed in more detail.

Response: We agree with the suggestions and added few lines in the introduction section too as per reference: doi: 10.4103/0973-7847.156314.  Line no. 66-70. However, readers can read suggested & cited paper for more information.

Comment 3: In the Figure 1, the authors present the basic skeleton of curcurbitacins, why the (B) CucurbitacinI is shown in the figure?

Response: We are thankful to reviewer for this in depth observation. However, there is no any specific reason to draw only cucurbitacinI and it is just to elaborate the difference between Cucurbitacin basic Skelton and other tetracyclic triterpenes. We further added atom numbering in the chemical structures for more clarity. Line no. 103

Round 2

Reviewer 1 Report

The authors revised the manuscript in accordance with the suggestions, and the quality of the manuscript has been improved. I think the manuscript can be now accepted for publication.